# Choclo virus (CHOV) recovered from deep metatranscriptomics of archived frozen tissues in natural history biorepositories

**Paris S. Salazar-Hamm** [1,2,3]*, **William L. Johnson**[4], **Robert A. Nofchissey**[2], **Jacqueline R. Salazar**[5], **Publio Gonzalez**[5], **Samuel M. Goodfellow**[2], **Jonathan L. Dunnum**[3,6], **Steven B. Bradfute**[2], **Blas Armién**[5,7], **Joseph A. Cook**[3,6], **Daryl B. Domman**[1,2], **Darrell L. Dinwiddie**[4]*

**1** Clinical and Translational Science Center, University of New Mexico, Albuquerque, New Mexico, United States of America, **2** Center for Global Health, Department of Internal Medicine, University of New Mexico Health Sciences Center, Albuquerque, New Mexico, United States of America, **3** Department of Biology, University of New Mexico, Albuquerque, New Mexico, United States of America, **4** Department of Pediatrics, University of New Mexico Health Sciences Center, Albuquerque, New Mexico, United States of America, **5** Department of Research in Emerging and Zoonotic Infectious Diseases, Gorgas Memorial Institute of Health Studies, Panama City, Panama, **6** Museum of Southwestern Biology, University of New Mexico, Albuquerque, New Mexico, United States of America, **7** Sistema Nacional de Investigación (SNI), Secretaria Nacional de Ciencia, Tecnología e Innovacion (SENACYT), Panama City, Panama

* psalazarhamm@salud.unm.edu (PSSH); DLDinwiddie@salud.unm.edu (DLD)

**Data Availability Statement:** All sequence data generated in this study were submitted to public archives. Reads were deposited in NCBI Short

## Abstract

### Background

Hantaviruses are negative-stranded RNA viruses that can sometimes cause severe disease in humans; however, they are maintained in mammalian host populations without causing harm. In Panama, sigmodontine rodents serve as hosts to transmissible hantaviruses. Due to natural and anthropogenic forces, these rodent populations are having increased contact with humans.

### Methods

We extracted RNA and performed Illumina deep metatranscriptomic sequencing on *Orthohantavirus* seropositive museum tissues from rodents. We acquired sequence reads mapping to Choclo virus (CHOV, *Orthohantavirus chocloense*) from heart and kidney tissue of a two-decade old frozen museum sample from a Costa Rican pygmy rice rat (*Oligoryzomys costaricensis*) collected in Panama. Reads mapped to the CHOV reference were assembled and then validated by visualization of the mapped reads against the assembly.

### Results

We recovered a 91% complete consensus sequence from a reference-guided assembly to CHOV with an average of 16X coverage. The S and M segments used in our phylogenetic analyses were nearly complete (98% and 99%, respectively). There were 1,199 ambiguous base calls of which 93% were present in the L segment. Our assembled genome varied 1.1% from the CHOV reference sequence resulting in eight nonsynonymous mutations.

Read Archive (SRA) under BioProject
PRJNA1015235. Host mitochondrial cytochrome b
sequence was deposited under GenBank accession
OR365535. The recovered Choclo virus assembly
was deposited in NCBI GenBank under accessions
OR365536, OR365537, OR365538. The above
sequence data was additionally linked to the
museum specimen voucher (https://arctos.
database.museum/guid/MSB:Mamm:131232).

**Funding:** This work has been partially supported by
the National Science Foundation PIPP Phase I
grant (NSF 2155222 to JAC). Continued clinical
and animal surveillance in Panama has been
supported by an Opportunity Pool Award
supplemented by the International Centers for
Infectious Diseases Research Program of the
National Institutes of Health (U19-AI 45452); funds
from the Gorgas Memorial Institute for Health
Studies, Hantavirus Research Project (04-90-0075-
8 to BA); funds from the Secretaria Nacional de
Ciencia y Tecnologia (Innovation and Technology
Program ftd06-089 to BA); funds from the Ministry
of Economy and Finance of Panama (FPI-MEF-056,
111130150.501.274, and PHoEZyTV I-II to BA).
The funders had no role in study design, data
collection and analysis, decision to publish, or
preparation of the manuscript.

**Competing interests:** The authors have declared
that no competing interests exist.

Further analysis of all publicly available partial S segment sequences support a clear relationship between CHOV clinical cases and *O. costaricensis* acquired strains.

## Conclusions

Viruses occurring at extremely low abundances can be recovered from deep metatranscriptomics of archival tissues housed in research natural history museum biorepositories. Our efforts resulted in the second CHOV genome publicly available. This genomic data is important for future surveillance and diagnostic tools as well as understanding the evolution and pathogenicity of CHOV.

## Author summary

Hantavirus cardiopulmonary syndrome (HCPS) in Panama, caused by Choclo virus (CHOV, *Orthohantavirus chocloense*), is intimately linked to the primary mammalian reservoir host, the Costa Rican pygmy rice rat (*Oligoryzomys costaricensis*). Although the prevalence of hantavirus disease is relatively low in Panama, over a quarter of the country has the agroecological conditions that favor this rodent. In addition, serologic evidence suggests infections are under-reported. Sequence data of the pathogen and host collected across temporal and spatial scales is necessary for diagnostics, surveillance, and forecasting; however, only one complete genome is available in NCBI GenBank. By leveraging deep metatranscriptomics of archived frozen mammal tissues, we generated a low-coverage genome using a reference-guided assembly approach. Sequence data can be used to develop pan-hantavirus diagnostic tools to facilitate acquisition of more detailed genetic data from archival samples to increase our understanding of the evolutionary and population dynamics of rare and neglected hantaviruses. This approach further illustrates the utility of cryopreserved biorepositories archived in natural history museums for pathogen discovery and pathobiology. Generating additional genomic sequence data will also be essential for developing a rigorous taxonomic framework to improve the understanding of hantavirus diversity and distribution.

## Introduction

Hantaviruses are tri-segmented negative-stranded RNA viruses within the family Hantaviridae that can cause two severe diseases in human populations, namely hemorrhagic fever with renal syndrome (HFRS) and hantavirus cardiopulmonary syndrome (HCPS). Hantaviruses in the Americas are more closely associated with HCPS, which is characterized by fever, headache, myalgia, hypotension, and thrombocytopenia that can progress to cardiopulmonary failure. The mortality rate for HCPS is estimated between 15–50% and varies among virus species and across countries [1–5].

Despite high mortality rates in humans, hantaviruses are maintained naturally in rodent populations and can persist for months to the lifetime of the animal [6]. Infected rodents shed virus through saliva, urine, and feces which can form aerosols that can be inhaled by other rodents or humans [7]. In Panama, the Costa Rican pygmy rice rat (*Oligoryzomys costaricensis*) serves as a rodent reservoir for Choclo virus (CHOV, *Orthohantavirus chocloense*). *O. costaricensis* is distributed across areas which are experiencing high levels of habitat

conversion from natural to agricultural lands. This transformation is hypothesized to increase population densities of commensal rodents and subsequent human contact with infected rodents, ultimately increasing zoonotic pathogen transmission [8]. CHOV was first identified by RT-PCR after an outbreak of HCPS in the agroecosystems of western Panama from December 1999 to February 2000 [9,10]. Monitoring of human and rodent populations in Panama over the past two decades have discovered multiple hantaviruses in Panama (e.g., Calabazo virus and Rio Segundo virus); however, CHOV is responsible for almost all human cases [11–13]. From 2001 to 2007 multiple community-wide surveys of western Panamanians without reported HCPS symptomology found 16–60% of the individuals were hantavirus seropositive depending on region [14], documenting that many mild or asymptomatic exposures were not accounted for in clinical case count data. CHOV-associated disease generally has a lower case fatality rate (mean 7.9%, range from 3% to 50.0%) than other New World Hantaviruses, but unfortunately predominantly affects young people (ages 20–49) [12].

Despite being first reported more than 40 years ago, hantaviruses are often described as 'emerging' pathogens due to their increasing number of infections, global distribution, and great breadth of pathogen diversity [15]. Current proactive approaches aimed at pathogen prediction are utilizing hantaviruses as a model for understanding spillover events [16]. Diverse specimens of wild mammals archived in museum biorepositories over temporal and spatial scales are increasingly being utilized for surveillance and characterization of emerging diseases [17–20]. For instance, the first complete CHOV sequence was obtained from archival *Oligoryzomys costaricensis* (= *fulvescens*, [21]) splenic tissue (MSB:Mamm:96073) using Sanger sequencing [13]. Of the 69 CHOV sequences available in NCBI GenBank, only nine are considered complete segments, eight of which are derived from the same voucher specimen (accessed August 10, 2023). Here, we deep sequenced archived mammalian tissue specimens to generate a metatranscriptome-assembled CHOV genome doubling the total available CHOV genomes.

## Methods

### Sample acquisition

We requested ten samples designated *Orthohantavirus* positive by an immunoglobulin G (IgG) serological screening test from the University of New Mexico Museum of Southwestern Biology (MSB) for metatranscriptomic sequencing. Upon collection of specimens in the field, tissues were immediately flash frozen in liquid nitrogen, subsequently transferred to -80˚C freezers at the Instituto Conmemorativo Gorgas in Panama City, and then permanently archived in vapor phase nitrogen freezers (-196˚C) in the MSB Division of Genomic Resources. One sample (10%), MSB:Mamm:131232, resulted in sufficient sequencing depth to assemble a Choclo virus (CHOV, *Orthohantavirus chocloense*) RNA genome. This voucher specimen was from an adult female Costa Rican pygmy rice rat (*Oligoryzomys costaricensis*) collected in El Bebedero, Tonosi, Los Santos, Panama in January 2003. Identification of host species was based on morphological characters and was subsequently verified using mitochondrial cytochrome b region sequence data (GenBank accession OR365535).

### RNA extraction, amplification, and sequencing

We performed a total RNA extraction on frozen tissue using the QIAamp viral RNA minikit (Qiagen Inc, Cambridge, MA, USA) according to the manufacturer's instructions, with slight modifications. Briefly, the tissue was bead beaten using 0.8 g of 1.0 mm Zirconia beads (BioSpec Inc, Bartlesville, OK, USA) and 1.5 g of 2.3-mm Zirconia beads (BioSpec Inc) in 800 μl of AVL buffer using the Benchmark Bead Bug-6 homogenizer at a speed of 4,350 rpm for 45 sec

for 2 cycles with a 1 min rest in between. Homogenates went through a series of centrifugation and transfers, first at 4,000 rpms for 7 min, then again at 7,000 rpms for 10 min to pellet debris. The clear lysate was transferred to a new tube with the RNA carrier and vortexed for 15 sec. The final RNA isolation was conducted per manufacturer's protocol including final elution with 50 µl nuclease free water.

Utilizing the Zymo RNA Clean & Concentrator-5 kit, extracted RNA was concentrated and treated with DNAse I utilizing on-column digestion following the manufacturer's protocol (Zymo Research, Irvine, CA, USA). The resulting RNA was depleted of ribosomal RNA for two hours, converted to cDNA using random hexamers, and i7 and i5 sequencing adaptors added. Finally, individual samples were barcoded and amplified by PCR (7 cycles). All depletion and library preparation steps were conducted using the Zymo-Seq RiboFree Total RNA Kit (Zymo Research) following the manufacturer's recommended protocol for degraded RNA. Prepared libraries were normalized to 2 nM, pooled, and combined with PhiX control (v.3, Illumina Inc, San Diego, CA, USA) at a final concentration of 1%. Pooled libraries were loaded at a final concentration of 750 pM and sequenced on an Illumina NextSeq 2000 using a P3 2x150 kit (Illumina Inc). De-multiplexing, adapter trimming, and preliminary QC were conducted using the Dragen pipeline (v.1.3.0, Illumina Inc). Reads were submitted to the NCBI Short Read Archive (SRA) under BioProject PRJNA1015235.

## Metatranscriptomic RNA genome assembly

We initially processed the sequence data with the nextflow nf-core/mag pipeline for assembly, binning, and annotation of metagenomes [22]. Initial k-mer based taxonomic classification was performed by Kraken2 [23]. To maximize recovery of CHOV reads, Illumina paired-end reads were mapped to the CHOV reference sequence using BWA v.0.7.17 [24]. The reference consisted of the S and M segments from the RefSeq assembly (NC_038373.1 and NC_038374.1) and a more recently sequenced L segment (KT983773.1) linked through the Arctos specimen record (https://arctos.database.museum/guid/MSB:Mamm:96073). Mapped paired-end reads were filtered and converted from bam to fastq files using SAMtools v.1.15.1 [25]. We generated an assembly from the mapped reads using SPAdes v.3.15.5 [26]. The assembly and mapped reads were then visualized with Artemis v.18.0.0 [27] for validating base calls. A threshold for trusted base calls was set at 3X coverage. Assembly quality was assessed with QUAST v.5.2.0 [28]. The resulting assembly for each segment was deposited in NCBI GenBank under accessions OR365536, OR365537, OR365538 and linked to the voucher specimen as best practice [19,29]. We used the Bacterial and Viral Bioinformatics Resource Center (BV-BRC) v.3.30.19a annotation tool for protein prediction (https://www.bv-brc.org/).

## Phylogenetic analyses

Reference sequences were obtained from GenBank for New World hantaviruses including *Orthohantavirus andesense* (Andes virus, ANDV), *Orthohantavirus bayoui* (Bayou virus, BAYV), *Orthohantavirus nigrorivense* (Black Creek Canal virus, BCCV), *Orthohantavirus chocloense* (Choclo virus, CHOV), *Orthohantavirus moroense* (El Moro Canyon virus, ELMCV), *Orthohantavirus negraense* (Laguna Negra virus, LANV and Rio Mamore virus, RIOMV), *Orthohantavirus montanoense* (isolate Limestone Canyon virus, LSCV), *Orthohantavirus sinnombreense* (New York virus, NYV and Sin Nombre virus, SNV), and for the Old World hantavirus *Orthohantavirus seoulense* (Seoul virus, SEOV) as an outgroup [30] (S1 Table). For each segment, sequences were aligned with mafft v.7.487 [31] using automatically determined settings (i.e., mafft—auto). The alignment was trimmed with trimal v.1.4.rev22 [32] in *automated1* mode with the additional removal of positions with <50% representation

(—resoverlap) and sequences with <60% representation (—seqoverlap). The resulting alignments were 1,825 bp for the S segment, 3,618 bp for the M segment, and 6,562 bp for the L segment. A concatenated alignment of the 5,443 bp complete S and M segment was also generated. Maximum likelihood trees were built in IQ-Tree v.1.6.12 [33] with the GTR +GAMMA model with 10,000 ultrafast bootstraps and 10,000 bootstraps for the SH-like approximate likelihood ratio (SH-aLRT) and visualized in ggtree [34] for each segment and the concatenated alignment. A tanglegram for evaluation of phylogenetic concordance between segments was visualized in R v.4.2.2.

Because the nucleocapsid protein is a primary detection marker for clinical diagnostics [35] and therefore the most abundant in sequence archives, we obtained 55 partial S segment sequences from GenBank by searching 'Choclo orthohantavirus' to explore strain diversity (S2 Table). The sequences were aligned, trimmed, and filtered as above resulting in a 585 bp alignment. A maximum likelihood was built under the GTR+GAMMA model in IQ-Tree with ultrafast and SH-aLRT bootstraps and visualized in ggtree as described above.

## Spatial distribution

We acquired GPS localities for capture sites of 32 Costa Rican pygmy rice rat (*O. costaricensis*) voucher specimens (https://arctos.database.museum) and for the approximate residency of 21 clinical cases of HCPS (S2 Table). The administrative political division of the Panamanian map was generated by the Topography Department of the National Institute of Statistics and Census of the General Comptroller of the Republic of Panama (https://www.inec.gob.pa/). A spatial representation of the human cases and rodent capture sites were georeferenced using the Datum UTM, WGS 1984, with ArcMap software from ArcGIS v.10.7 (ESRI2019) [36]. Samples collected within 12 km were aggregated based on maximum spatial movements of close rodent relatives [37,38]. Although we only present samples here with sequence data, previous surveillance efforts have found hantavirus seropositive *O. costaricensis* across its broad geographic range in Panama in five out of the nine ecoregions: Central American Atlantic Moist Forests, Isthmian-Pacific Moist Forests, Panamanian Dry Forests, Pacific Mangrove S. America, and Choco/Darién Moist Forests [36].

## Results and Discussion

An outbreak from late 1999 to early 2000 of hantavirus cardiopulmonary syndrome (HCPS) in western Panama [9,10] has spurred over two decades of epidemiological and wildlife surveillance [12,36]. During those 20 years 712 clinical cases of HCPS were reported in Panama [12] and >11,000 specimens of non-volant mammals with archived biological materials were contributed to museum repositories (https://arctos.database.museum/). Of these, 883 have been identified as *O. costaricensis*, and 778 (88%) have been screened for prior hantavirus infection using an IgG strip immunoblot assay [39] with an average seropositivity of 16% [36]. We requested samples designated hantavirus positive at the University of New Mexico MSB for metatranscriptomic sequencing. Initial taxonomic assignment by Kraken2 classified < 15 reads *Orthohantavirus* in all samples except MSB:Mamm:131232, for which it identified 658 reads. Hantavirus-specific IgG antibodies can be detected up to six months after initial infection [40]. Given that 90% of the hantavirus-positive samples had 0 to 14 reads, there was likely no active infection.

We then took a reference-based mapping approach to maximize recovery of CHOV reads. We generated 113,237,341 reads from deep sequencing the metatranscriptome of MSB: Mamm:131232, of which 2,072 reads (0.00002%) mapped to the CHOV reference. The average depth of coverage from the mapping-based alignment was 16X; however, there was variation

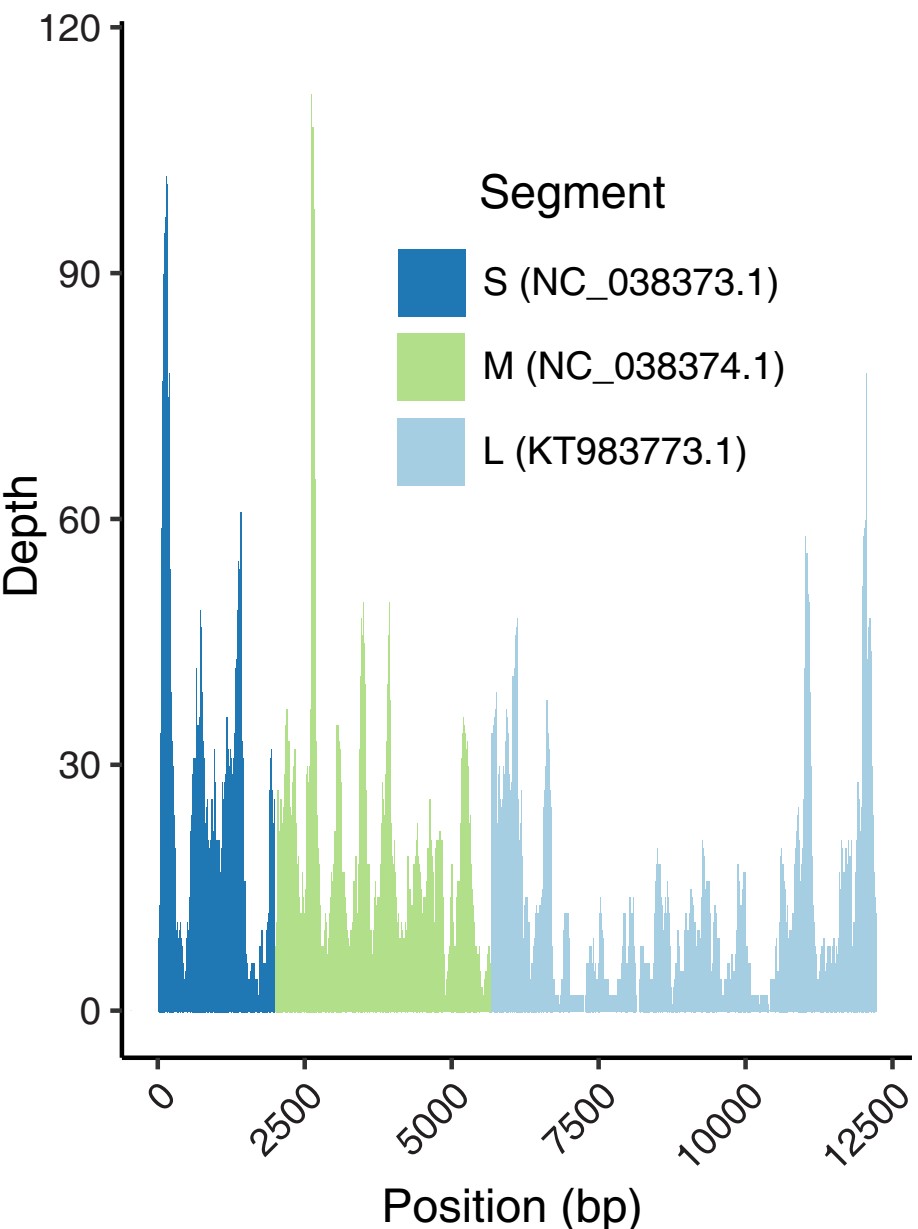

**Fig 1. Coverage plot of sequencing reads mapped to the S, M, and L segments of the CHOV reference genome.**

between segments with the greatest coverage of the small (S) segment (25X) followed by the medium (M) segment (19X) and the large (L) segment (12X) (Fig 1). Due to the low abundance of sequencing reads mapping to CHOV, we implemented a 3X coverage threshold of base-calling in the consensus genome. If the coverage at a site was below 5X, we only called a base when the allele frequency was 100% at that site. Using these parameters, we recovered a 91% complete CHOV assembly. Similar coverage thresholds have aided in the recovery of more complete assemblies [41,42]. This was an improvement upon our initial *de novo* assembly which was only 64% complete. Under these thresholds, there were 1,199 ambiguous base calls of which 93% (1,114 bp) were in the L segment. Therefore, completeness of the L segment

(83%) was less than the M (99%) and the S (98%) segments. While our L segment is not complete, this marks a major improvement compared to many of the hantaviruses deposited in GenBank which are missing the L segment (ELMCV, LANV, and NYV) or only sequenced small portions of the L segment (BCCV). The International Committee on Taxonomy of Viruses Hantaviridae Study Group is revisiting requirements for proper hantavirid classification including standards for minimum sequence quality and genome completeness [43] as well as minimum information standards for uncultured viral genomes [44]. Limited biological material precluded additional sequencing necessary for genome completion and verification of low coverage base-calls. However, our efforts are still highly valuable as they provide the second CHOV, which will aid in our ability to design PCR amplicon tiling array and hybridization target capture sequencing strategies that can lead to affordable and scalable targeted sequencing efforts [45]. Such protocols that enable high throughput and cost-effective means to generate robust genetic information are needed for future genomic comparative studies and systematic exploration of understudied viruses, such as CHOV.

To place our CHOV genome into the phylogenetic context of other hantaviruses, we performed single and concatenated segment alignments and generated maximum likelihood phylogenetic trees. Our sequence fell into a well-supported clade (≥99% bootstrap values) with the CHOV reference genome from voucher MSB:Mamm:96073 (Figs 2A and S1). Both the reference and our sequence were isolated from *O. costaricensis* specimens from Los Santos, Panama, and were captured 23 months apart (Fig 3). Excluding ambiguous base calls (Ns), there are 125 single nucleotide polymorphisms (SNPs) differing from the CHOV reference of which 40 SNPs were in the L segment, 58 SNPs were in the M segment, and 27 SNPs were in the S segment. This corresponds to a divergence of 1.1% from the CHOV reference genome. This variation resulted in eight non-synonymous mutations with five located in the M segment (p. D246G, p.S247F, p.K510R, p.A658T, and p.V1133G) and three in the L segment (p.L342S, p. V412I, and p.N778S) (S3 Table). Overall, the limited number of amino acid differences discovered across two samples collected nearly a year apart, but within the same geographic region is interesting. However, to understand the true extent of genomic heterogeneity of CHOV regionally and temporally, generating additional complete genomes from both rodents and humans are necessary. It should be noted, that while the CHOV reference genome was generated with conventional Sanger sequencing (average accuracies of ~99.4%), Illumina next-generation sequencing (NGS) produces comparable sequence accuracies for reads with phred-like quality scores above 30, and overall accuracy increased with increasing depth of coverage [46]. The M segment encodes for two glycoproteins (G1 and G2), involved in signal transduction and mitigation of host antiviral defenses, positions 22–545 and 658–1136, respectively. While amino acid substitutions from lysine (K) to arginine (R), or alanine (A) to threonine (T) are less likely to affect protein structure because of similar side chain polarity and charge, modifications from aspartic acid (D) or valine (V) to glycine (G) are more likely to cause structural changes because of differing side chain biochemistry. Structural and functional studies of the CHOV L segment, similar to those in SNV [47], should be investigated. Non-synonymous mutations can have effects on protein stability and structure or alter protein-protein interactions [48]. In addition, variation in the non-coding regions at the 5′ and 3′ termini could disrupt complementary sequences which aid in the panhandle structure formation essential for transcription and replication [49]. These regions are generally highly conserved, and we observed only one SNP in the S segment (g.T29C), which is not predicted to interfere with panhandle formation.

Hantavirus evolution has likely been shaped by their co-evolutionary history with their rodent reservoirs [50]. The phylogenetic relationships between South American and North American hantaviruses from sigmodontine rodents (Fig 2B) are most likely derived from a

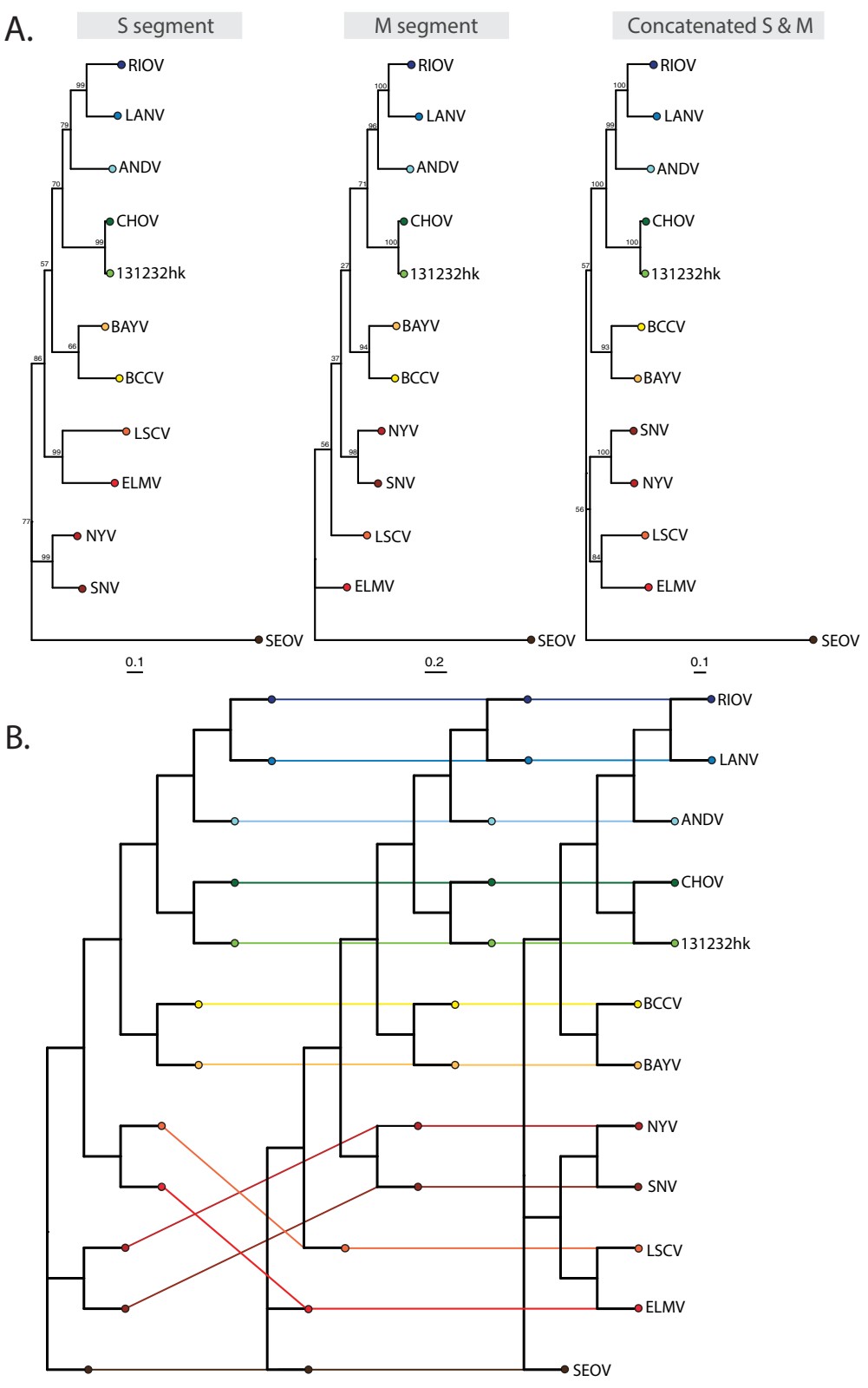

**Fig 2. Maximum likelihood phylogenies of the S and M segments and a concatenated alignment (A) and a tanglegram used to visualize possible reassortment histories (B).** Phylogenies were built with the GTR+GAMMA model with 10,000 ultrafast bootstraps and 10,000 bootstraps for the SH-aLRT on single segment and concatenated alignments. In the tanglegram, lines connect the same taxa/tip in each tree to one another such that crossing lines suggest topological discordance.

complex history of co-speciation events and the biogeographic constraints that influenced rodent expansion into South America [51]. However, intra- and inter- lineage reassortment between closely related variants have been reported for several hantaviruses [52–56]. Although assumptions of reassortment events are often based on conflicting phylogenetic tree topologies, reassortment has also been demonstrated in *in vitro* experiments [57–59]. The high degree of genomic similarity in the S and L segments suggests the exchange of the M segment is more common and potentially beneficial [60]. We did not find evidence of reassortment between the two CHOV genomes (Fig 2B). With additional genomic variants, this question should be revisited with more robust analyses [61].

One outstanding question is the relatedness of hantaviruses isolated from rodents to human clinical cases. A phylogenetic analysis of all publicly available partial S segment CHOV sequences clearly demonstrates viruses isolated from *O. costaricensis*, including the strain described here, are intimately related to all clinical strains from Panama (Fig 4). Both clinical

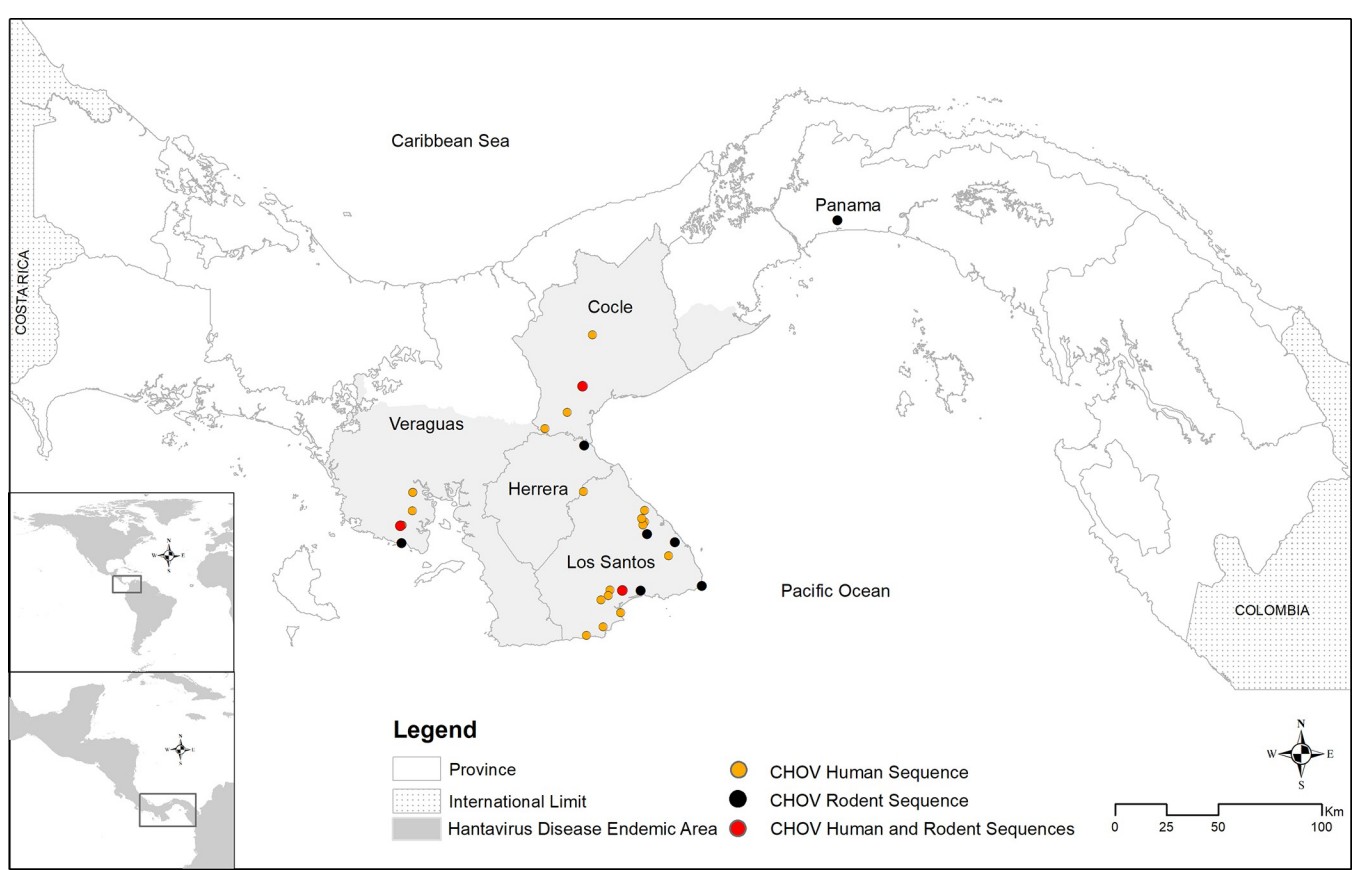

**Fig 3. Geographic distribution of Panamanian CHOV sequences from 32 capture sites of the Costa Rican pygmy rice rat (O. costaricensis) and approximate residence of 21 clinical cases georeferenced using the Datum UTM, WGS 1984, with ArcMap software from ArcGIS (ESRI2019).** Red dots equating to the overlap in the locality of human and rodent sequences represent six rodents and one human in Los Santos, two rodents and two humans in Veraguas, and two rodents and one human in Coclé.

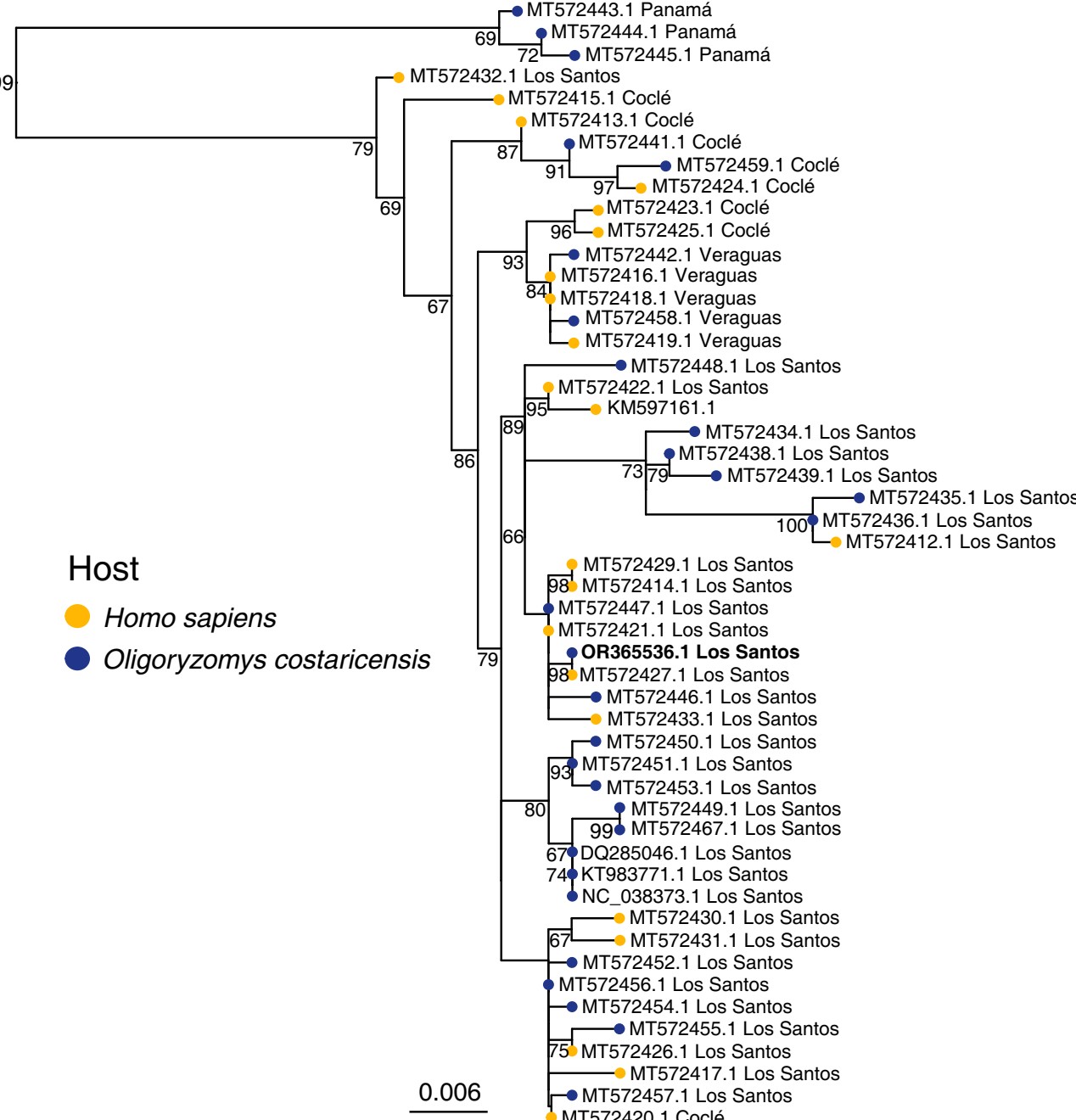

**Fig 4. A maximum likelihood phylogeny of the partial S segment of 51 CHOV genomes demonstrates sequences isolated from *O. costaricensi*s are associated with human disease.** The phylogeny was built with the GTR+GAMMA model with 10,000 ultrafast bootstraps and 10,000 bootstraps for the SH-aLRT. Bootstrap values ≥ 65% are shown. The sequence in bold was acquired in this study.

and rodent CHOV strains were captured from Los Santos, Veraguas, and Coclé provinces; however, only rodent-acquired sequences were found in Panamá (Fig 3). This is reflective of the clinical disease burden with the greatest number of cases in Los Santos (77%) followed by Veraguas (12%) and Coclé (7%) [12]. However, it should be noted that locations of clinical cases are by residence, and it is possible patients were exposed while traveling to another

endemic area of the country. The strains from the province of Panamá form a well-supported clade demonstrating potential geographic substructure (Fig 4) which is congruent with previous findings [36]. More sequencing is needed to determine how closely these are related to clinical strains; however, only seven Panamá residents have reported HCPS in the last 20 years (1% of all cases) [12]. Five other sequences from Panamanian hantaviruses were isolated from the short-tailed cane mouse (*Zygodontomys brevicauda*) and the Chiriqui harvest mouse (*Reithrodontomys creper*), but these are likely representative of other unclassified hantaviruses (e.g., Calabazo virus and Rio Segundo virus) that have yet to be fully sequenced [43] or associated with human disease (S2 Fig).

Virus discovery has been dramatically accelerated with the advances in sequencing [62]. Here, we provide a proof-of-concept for using deep metatranscriptomics to recover a hantavirus genome from archived museum tissue samples. Viral enrichment protocols prior to sequencing, including those that utilize fraction and filtration methods, may even increase recovery of target organisms [63,64]. Other methods, such as PCR or target enrichment by capture [45], that seek to enrich virus sequences during library prep, require a priori knowledge of genomic sequences to preform optimally, and will be assisted by the metatranscriptomic sequencing described here. Furthermore, we demonstrate that tissues from voucher MSB:Mamm:131232, which were ultrafrozen for 20 years before our current sequencing efforts, were still viable for NGS, highlighting the longevity and utility of such preserved specimens. Our understanding of hantavirus systematics, evolution, and ecology have been profoundly influenced by integrating collaborative efforts between public health agencies, virologists, field biologists, and museum scientists. Museum specimen collections have been leveraged to document the distribution of SNV [65] and the discovery of multiple novel hantaviruses [66]. Utilizing these invaluable archives is essential to understanding pathogens circulating past and present. Given that habitat conversion [8] and climate change [67] could influence reservoir host densities and therefore increase the burden of CHOV and other hantaviruses, it is necessary to continue capturing sequence data to inform diagnostics, detection, and vaccine design.

## Supporting information

**S1 Table. *Orthohantavirus* sequences obtained from GenBank.**
(XLSX)

**S2 Table. Partial S segment of 56 Panamanian *Orthohantavirus* sequences obtained from GenBank.**
(XLSX)

**S3 Table. Nonsynonymous mutations against the CHOV reference.**
(XLSX)

**S1 Fig. A concatenated 11,972 bp alignment of the S, M, and L segments was used to infer a maximum likelihood phylogeny of New World Hantaviruses.** The phylogeny was built with the GTR+GAMMA model with 10,000 ultrafast bootstraps and 10,000 bootstraps for the SH-aLRT. LANV, BCCV, ELMCV, and NYV were limited to just the complete S and M segments.
(PDF)

**S2 Fig. A maximum likelihood phylogeny of the partial S segment of 56 Panamanian hantaviruses.** Sequences were obtained from sick human patients and three rodent hosts (*Oligoryzomys costaricensis*, *Zygodontomys brevicauda*, and *Reithrodontomys creper*).
(PDF)

## Acknowledgments

The authors thank the field crews and museum scientists at the Instituto Conmemorativo Gorgas de Estudios de la Salud (Panama City) and Museum of Southwestern Biology (Albuquerque) for collecting and preserving archival tissues of the rodents. Specifically, we thank the Panamanian Ministry of Environmental Affairs, the Panamanian Ministry of Health, the Panamanian Institute of Livestock and Agricultural Research, the Gorgas Committee for Animal Care and Use, and the International Center for Infectious Disease Research Program of the National Institutes of Health for their support. We also thank the UNM Center for Advanced Research Computing for providing high performance computing resources.

## Author Contributions

**Conceptualization:** Daryl B. Domman, Darrell L. Dinwiddie.

**Data curation:** Paris S. Salazar-Hamm, William L. Johnson, Robert A. Nofchissey, Jacqueline R. Salazar, Publio Gonzalez, Jonathan L. Dunnum, Blas Armién, Joseph A. Cook, Darrell L. Dinwiddie.

**Formal analysis:** Paris S. Salazar-Hamm, Samuel M. Goodfellow.

**Investigation:** Blas Armién, Joseph A. Cook, Daryl B. Domman.

**Methodology:** Paris S. Salazar-Hamm, William L. Johnson, Daryl B. Domman, Darrell L. Dinwiddie.

**Supervision:** Steven B. Bradfute, Blas Armién, Joseph A. Cook, Daryl B. Domman, Darrell L. Dinwiddie.

**Writing – original draft:** Paris S. Salazar-Hamm.

**Writing – review & editing:** Paris S. Salazar-Hamm, William L. Johnson, Robert A. Nofchissey, Jacqueline R. Salazar, Publio Gonzalez, Jonathan L. Dunnum, Steven B. Bradfute, Blas Armién, Joseph A. Cook, Daryl B. Domman, Darrell L. Dinwiddie.

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
