## [Decision Letter · Decision Letter 0]

16 Nov 2023

Dear Dr. Salazar-Hamm,

Thank you very much for submitting your manuscript "Choclo virus (CHOV) recovered from deep metatranscriptomics of archived museum tissues" for consideration at PLOS Neglected Tropical Diseases. As with all papers reviewed by the journal, your manuscript was reviewed by members of the editorial board and by several independent reviewers. The reviewers appreciated the attention to an important topic. Based on the reviews, we are likely to accept this manuscript for publication, providing that you modify the manuscript according to the review recommendations. 

All reviewers comment on the methods, and the authors should provide justification regard the specific methodological issues raised, both in the responses and in the manuscript (e.g., acknowledging the limitations of the methods/approach). Most of these comments could be easily addressed by adding the requested information. Reviewers 1 & 2 suggest additional experiments, but both acknowledge that these might not be possible and are therefore not required for reconsideration of the revised manuscript, although the points are valid and the authors could of course receive extra time in generating their revision to address these issues. If they cannot perform some of the additional experiments, the revised manuscript should address these issues directly. Specifically concerning the level of confidence in the called nucleotides from a metagenomic assembly with relatively low coverage and depth, and stating why additional measures were not taken to give more support to the accuracy of the resulting genomic sequence(s) (e.g., enrichment or validation with amplicon sequencing via Sanger or NGS).

Reviewer 3 suggests including an inset map of the region on a larger scale, which I agree is essential.

Sincerely,

Jeremy V. Camp, Ph.D.

Academic Editor

Andrea Marzi

Section Editor

All reviewers comment on the methods, and the authors should provide justification regard the specific methodological issues raised, both in the responses and in the manuscript (e.g., acknowledging the limitations of the methods/approach). Most of these comments could be easily addressed by adding the requested information. Reviewers 1 & 2 suggest additional experiments, but both acknowledge that these might not be possible and are therefore not required for reconsideration of the revised manuscript, although the points are valid and the authors could of course receive extra time in generating their revision to address these issues. If they cannot perform some of the additional experiments, the revised manuscript should address these issues directly. Specifically concerning the level of confidence in the called nucleotides from a metagenomic assembly with relatively low coverage and depth, and stating why additional measures were not taken to give more support to the accuracy of the resulting genomic sequence(s) (e.g., enrichment or validation with amplicon sequencing via Sanger or NGS).

Reviewer 3 suggests including an inset map of the region on a larger scale, which I agree is essential.

Reviewer's Responses to Questions

**Key Review Criteria Required for Acceptance?**

**Methods**

-Are the objectives of the study clearly articulated with a clear testable hypothesis stated?

-Is the study design appropriate to address the stated objectives?

-Is the population clearly described and appropriate for the hypothesis being tested?

-Is the sample size sufficient to ensure adequate power to address the hypothesis being tested?

-Were correct statistical analysis used to support conclusions?

-Are there concerns about ethical or regulatory requirements being met?

Reviewer #1: (No Response)

Reviewer #2: (No Response)

Reviewer #3: This paper describes the generation of a quasi-complete genome of the Choclo virus from tissues obtained from the Museum's collection. The design of the study contributes to obtaining a low number (only one) of the sequences wanted; the difficulty of preserving the samples for years and the low number of hantavirus positive rodents, go against the probability of obtaining complete genomes, which leads to having a low number of specimens to give greater statistical validity and more likely to achieve the desired objective. In the bioinformatics analysis, measures were taken to avoid confusion between the errors inherent in the technique and the real variability of the genome. The authors should mention the different error rate in both sequencing techniques (Sanger, for sequences published in the past, and NGS for new sequences), when comparing both complete Choclo genomes. Knowing the difficulty of generating complete sequences of hantavirids in general, the study design could have given better results if a method of enrichment of the target sequences had been included. Regarding the study of HCPS cases, in order to geolocalize the cases, it is important to consider the possibility of travel (within the incubation period of the patient) to another endemic area of the country.

**Results**

-Does the analysis presented match the analysis plan?

-Are the results clearly and completely presented?

-Are the figures (Tables, Images) of sufficient quality for clarity?

Reviewer #1: (No Response)

Reviewer #2: (No Response)

Reviewer #3: The results are in accordance with the methodology and the objective that is intuited in the approach of the work. The authors report preliminary results with Kraken2, which are not covered in the materials and methods section (in case it is part of this work) or cited in case it is the result of past work. A brief discussion of the low variability over time, and not only geographically, could be considered interesting, as specimens were sequenced from different years than previously published. In figure 3, contextualisation on a map, e.g. of America, would facilitate the location of the study region; it would also be desirable to have the scaling. In FigS1_SML_concatenated_phylo: is Maporal virus divergence correct? (MAPV sequence is published in PubMed on the contrary sense).

**Conclusions**

-Are the conclusions supported by the data presented?

-Are the limitations of analysis clearly described?

-Do the authors discuss how these data can be helpful to advance our understanding of the topic under study?

-Is public health relevance addressed?

Reviewer #1: (No Response)

Reviewer #2: (No Response)

Reviewer #3: The conclusions have been presented together with the results and reflect the limitations (low number of sequences obtained, missing sequenced fragments to complete the L-segment, etc.). The authors make clear three main issues for the study of CHOV and the prevention of HCPS in Panama: they provide one more complete genome of CHOV, the only one published so far; they provide a significant number of partial sequences of the S-segment that will allow further studies of variability in the region; and they present evidence of the close relationship between the viruses obtained from samples from humans and those from rodents.

**Editorial and Data Presentation Modifications?**

Reviewer #1: (No Response)

Reviewer #2: (No Response)

Reviewer #3: --

**Summary and General Comments**

Reviewer #1: Salazar-Hamm and colleagues have presented a valuable study on the descriptive evolutionary analysis of the Choclo virus. The analytical methods are well-executed, and the figures provided are informative. The overall structure of the article aligns with the general requirements of the journal. While I found the results regarding this hantavirus intriguing, I have noted some areas that need clarification concerning the sample processing and sequencing, as detailed below. Additionally, the connection of this work with general museum collections and voucher specimens appears misleading. With substantial revisions, I believe this work can be suitable for publication in the journal.

The term "archived museum samples" might be misleading, as standard museum storage typically involves preservation in liquid (formaline or alcohol) or dried form. I recommend changing the title to "frozen samples" and elaborating on the museum's relevance in the text. I apologize for the simplicity of this suggestion, but technically, these samples are stored under frozen conditions, which is a common method in laboratory settings. While I acknowledge the importance of highlighting the value of museum collections, please note that frozen storage is not the primary method employed in museum collections.

Please provide more details about the nature of the tissue samples. This information is crucial for understanding hantavirus tissue tropism and will be valuable for other research endeavors.

Sequencing coverage at several genomic positions is low, I miss the verification of these positions with other sequencing methods (amplicon-based NGS sequencing or Sanger sequencing).

Why didn't the authors perform any viral enrichment experiments before nucleic acid extraction? This could significantly enhance the yield of viral reads.

I dont expect the authors to conduct additional experiment – I understand the limitations of processing valuable samples, but I think the clarification of the lack of these experiments of giving more details about the sample processing would be beneficial for the readers.

Reviewer #2: The Costa Rican pygmy rice rat (Oligoryzomys costaricensis) serves as the main wildlife host for Choclo orthohantavirus (CHOV), which is responsible for hantavirus cardiopulmonary syndrome in humans in Panama. The primary objective of this study was to establish the feasibility of obtaining genomic sequences of CHOV from archival frozen rat tissues using a deep metatranscriptomics approach. This endeavor deserves recognition for two significant reasons. Firstly, there exists only one complete genome of this virus in the NCBI GenBank database. Secondly, the pursuit is equally commendable due to the scarcity of L segment sequences, despite it being one of the most conserved segments among hantaviruses, which are often absent in many hantavirus species or virus strains. 

In a prior study (PMID: 37376689), 778 samples from Costa Rican pygmy rice rats were screened for previous hantavirus infections using an IgG strip immunoblot assay, revealing that 16% of these samples (122) had tested positive. Some of these seropositive samples were further examined in the current study to detect the presence of CHOV genomic sequences through a transcriptomics approach. In the initial screening, only one sample produced a modest number of sequencing reads (>14), and was therefore subjected to deep sequencing, which generated a total of 113,237,341 sequencing reads. These reads were then assembled to produce a near-complete genome of the CHOV isolate. The sequences obtained were phylogenetically analyzed together with other publicly available CHOV sequences and other hantaviruses. 

A notable limitation of this study is that, despite extensive efforts, only a partial CHOV genome was obtained. Approximately 1,200 nucleotides remained ambiguous, accounting for roughly 10% of the entire virus genome. Notably, 95% of these ambiguous nucleotides originate from the largest genomic segment of CHOV, segment L. Consequently, while this sequence may find utility in subsequent PCR or sequencing investigations, its utility for phylogenetic studies is very limited. 

If the authors still possess RNA from the sample MSB:Mamm:131232 which contained a substantial abundance of CHOV RNA copies, it raises the question of whether it is feasible to obtain a complete or more complete sequence of the L segment. This could potentially be achieved through either PCR or a hybridization-based enrichment and sequencing approach. 

Another limitation of this manuscript pertains to the notion that, while the concept of extracting sequence data from archival samples holds promise, it lacks novelty. Ultimately, the manuscript provides a limited amount of new information, essentially detailing the acquisition of two near-complete S and M segment sequences of CHOV from an archival sample.

Minor comments

The manuscript does not explicitly specify the number of hantavirus-positive samples that underwent RNA extraction and screening through a sequencing approach.

The authors have provided descriptions of sequence differences in the protein coding regions between the obtained sequences and the reference sequences. However, they have not addressed differences in the non-coding regions. It would be valuable if the authors could also comment on the characteristics of the 5’ and 3’ termini of the obtained sequences. Do they show conserved structure, and have the capacity to form a panhandle structure?

The statement that outlines the intentions of the International Committee on Taxonomy of Viruses Hantaviridae Study Group to address the classification problems within the Hantaviridae family is very vague (lines 238-239). The authors should offer a more comprehensive description of the proposed reorganization and clarify the significance of obtaining complete genomes within this specific context. 

While it can be inferred that Figure 3 displays the map of Panama, this fact is not explicitly stated in the figure legend. Additionally, as the authors discuss the distribution of CHOV in various provinces within Panama, it would be beneficial for the map to indicate the locations of these provinces in Panama for enhanced clarity. 

The legend of Figure 4 should specify that the sequence labeled in bold font was acquired in the current study. 

Typo in line 137.

Reviewer #3: The work represents an innovative source of genetic information about viruses and uses modern technological means to extract this information. They do not achieve the goal of having several complete genomes, but they do obtain at least one and also several partial sequences of the s segment of the Choclo virus. The authors' proposed link between rodent information and cases is interesting, as this information can be used to improve the construction of risk maps for HCPS. Further progress will be needed to obtain a larger number of sequences in order to be able to reach the tentative conclusion presented by the authors on the possibility of reassociation of viral segments or genetic and/or amino acid modifications that lead to the alteration of a biological property. The work has shed light on the hantavirus situation in Panama.

PLOS authors have the option to publish the peer review history of their article (what does this mean?). If published, this will include your full peer review and any attached files.

Reviewer #1: No

Reviewer #2: No

Reviewer #3: No

Figure Files:

Data Requirements:

Reproducibility:

References

---

## [Decision Letter · Decision Letter 1]

3 Jan 2024

Dear Dr. Salazar-Hamm,

We are pleased to inform you that your manuscript 'Choclo virus (CHOV) recovered from deep metatranscriptomics of archived frozen tissues in natural history biorepositories' has been provisionally accepted for publication in PLOS Neglected Tropical Diseases.

Best regards,

Jeremy V. Camp, Ph.D.

Academic Editor

Andrea Marzi

Section Editor

Reviewer's Responses to Questions

**Key Review Criteria Required for Acceptance?**

**Methods**

-Are the objectives of the study clearly articulated with a clear testable hypothesis stated?

-Is the study design appropriate to address the stated objectives?

-Is the population clearly described and appropriate for the hypothesis being tested?

-Is the sample size sufficient to ensure adequate power to address the hypothesis being tested?

-Were correct statistical analysis used to support conclusions?

-Are there concerns about ethical or regulatory requirements being met?

Reviewer #1: (No Response)

Reviewer #2: (No Response)

Reviewer #3: With regard to this new revision (the second one), I have been able to appreciate the changes introduced by the authors, particularly in line 112 (Introduction section), where the information on the previously known sequences of CHOV has been extended.

In the Methods section of this new version, I noticed changes in line 123, where more information on the preservation of museum specimens has been added, as requested in the revision; and a new map and map data (line 207) with detailed context.

There are no new considerations.

**Results**

-Does the analysis presented match the analysis plan?

-Are the results clearly and completely presented?

-Are the figures (Tables, Images) of sufficient quality for clarity?

Reviewer #1: (No Response)

Reviewer #2: (No Response)

Reviewer #3: With respect to this new revision (the second), the present version of the manuscript includes positive changes:

* An explanation of why a sample enrichment step prior to sequencing was not included (line 250).

* Line 268 presents new numbers of SNPs, reducing the number of non-synonymous mutations.

* The new, more complete map allows discussion of the distribution of rodents and positive cases, including observations or comments made by the reviewers in the first peer review.

**Conclusions**

-Are the conclusions supported by the data presented?

-Are the limitations of analysis clearly described?

-Do the authors discuss how these data can be helpful to advance our understanding of the topic under study?

-Is public health relevance addressed?

Reviewer #1: (No Response)

Reviewer #2: (No Response)

Reviewer #3: (See results camp)

**Editorial and Data Presentation Modifications?**

Reviewer #1: (No Response)

Reviewer #2: (No Response)

Reviewer #3: Accept.

**Summary and General Comments**

Reviewer #1: All requested changes were implemented and all comments were addressed.

Reviewer #2: I thank the authors for addressing my concerns, especially for the discussion on the possibility of obtaining a more comprehensive viral sequence through additional experiments. As all concerns that I had raised have been satisfactorily addressed in the revised manuscript, I recommend it for publication.

Reviewer #3: In my opinion, with the changes and additional information, the manuscript is ready to continue the publication process.

PLOS authors have the option to publish the peer review history of their article (what does this mean?). If published, this will include your full peer review and any attached files.

Reviewer #1: No

Reviewer #2: No

Reviewer #3: No

---

## [Editor Report · Acceptance letter]

9 Jan 2024

Dear Dr. Salazar-Hamm,

We are delighted to inform you that your manuscript, "Choclo virus (CHOV) recovered from deep metatranscriptomics of archived frozen tissues in natural history biorepositories," has been formally accepted for publication in PLOS Neglected Tropical Diseases.

Best regards,

Shaden Kamhawi

co-Editor-in-Chief

Paul Brindley

co-Editor-in-Chief
